# Molecular Basis of Surgical Coaptation Techniques in Peripheral Nerve Injuries

**DOI:** 10.3390/jcm12041555

**Published:** 2023-02-16

**Authors:** Clifford T. Pereira, Elise E. Hill, Anastasiya Stasyuk, Neil Parikh, Jannat Dhillon, Aijun Wang, Andrew Li

**Affiliations:** 1Department of Surgery, University of California Davis Medical Center, Sacramento, CA 95817, USA; 2Division of Plastic Surgery, University of California Davis Medical Center, Sacramento, CA 95817, USA; 3Department of Surgery, David Grant Medical Center, Travis Air Force Base, Fairfield, CA 94535, USA; 4School of Medicine, University of California Davis, Sacramento, CA 95817, USA; 5School of Medicine, Boston University, Boston, MA 02118, USA; 6Boston University, Boston, MA 02118, USA

**Keywords:** peripheral nerve injury, surgical repair, nerve coaptation, molecular mechanisms, end-to-end coaptation, end-to-side coaptation, side-to-side coaptation, nerve regeneration

## Abstract

Peripheral nerve injuries requiring surgical repair affect over 100,000 individuals in the US annually. Three accepted methods of peripheral repair include end-to-end, end-to-side, and side-to-side neurorrhaphy, each with its own set of indications. While it remains important to understand the specific circumstances in which each method is employed, a deeper understanding of the molecular mechanisms underlying the repair can add to the surgeon’s decision-making algorithm when considering each technique, as well as help decide nuances in technique such as the need for making epineurial versus perineurial windows, length and dept of the nerve window, and distance from target muscle. In addition, a thorough knowledge of individual factors that are active in a particular repair can help guide research into adjunct therapies. This paper serves to summarize the similarities and divergences of the three commonly used nerve repair strategies and the scope of molecular mechanisms and signal transduction pathways in nerve regeneration as well as to identify the gaps in knowledge that should be addressed if we are to improve clinical outcomes in our patients.

## 1. Introduction 

Peripheral nerve injuries (PNIs) affect approximately twenty million people in the US with a combined 100,000 patients requiring peripheral nerve surgery in the US annually [1,2]. Peripheral nerve injuries are commonly caused by motor vehicle accidents and penetrating injuries, frequently involving working-age adults under 40, imparting a significant work force burden [3,4]. The current surgical gold standard is a direct end-to-end (ETE) repair; however, this approach is only viable if the resulting gap is small and amenable to a tension-free repair [1,2,5]. Historically, the most common direct repair technique is end-to-end coaptation, indicated when nerve segments can be approximated without tension in the presence of a viable proximal nerve stump [6,7,8,9]. In cases where direct repair is not feasible, autografts and/or cadaveric allografts are often employed to fill the gap. Coaptation of the nerve graft with the proximal and distal nerve stump is similarly done in and ETE fashion. In the absence of a proximal nerve stump, such as in nerve root avulsions, other techniques such as end-to-side repair or less frequently side-to-side repair is utilized to avoid the need for donor nerve sacrifice [6]. Multiple studies comparing the clinical outcomes of these procedures have been conducted and reviewed, however the molecular mechanisms behind each technique have not been reviewed before. This review aims to augment the classical views on nerve regeneration after end-to-end nerve coaptation with recently emerging data regarding less frequently utilized surgical techniques such as the end-to-side and side-to-side coaptation. We additionally aim to identify knowledge gaps and new research avenues for nerve coaptation research. 

## 2. Molecular Events Preceding Nerve Regeneration

Prior to discussing nerve coaptation techniques and the molecular mechanisms behind each of these, it is important to consider the general events occurring after nerve damage. Peripheral nerves consist of bundles of axons sheathed in three layers of connective tissue: epineurium, perineurium, and endoneurium (Figure 1). Each axon is enveloped by the endoneurium and Schwann Cells (SCs), the main glial cells of the peripheral nervous system. Groups of axons are organized into a fascicle by perineurium, and these fascicles are finally sheathed in epineurium to form the peripheral nerve. According to the Sunderland classification, peripheral nerve injuries are classified into five degrees depending on the mechanism and intensity of injury. These include segmental demyelination (first degree); axonal damage with intact endoneurium (second degree); axonal damage with endoneurium damage (third degree); axonal, endoneurial and perineurial damage (fourth degree); and loss of continuity of entire nerve trunk (fifth degree) [1,2]. For the purposes of this paper we will be discussing fifth degree nerve injury and regeneration. Damage to the axonal membrane results in two distinct temporal phases, a rapid phase consisting of rapid influx of sodium and calcium, and a slow phase characterized by activation and the retrograde transport of signaling molecules. The rapid influx of sodium and calcium activates calpain, leading to the sealing of the injured nerve membranes. It also generates a bust of action potentials at the proximal stump that is propagated to the soma via L-type voltage-gated calcium channels and the release of calcium from the endoplasmic reticulum. This results in the activation of the second messenger cAMP, which activates several important pro-regenerative transcription factors including dual leucine zipper-bearing kinase (DKL; also known as MAP3K12) and CREB1 (cAMP-responsive element-binding protein 1) [7,8,9,10]. The slower phase of injury is mediated via protein kinases, including calcium/calmodulin-dependent kinase 2(CMAK2); mitogen-activated protein kinases (MAPK) such as Erk1 and Erk2; phosphatidylinositol 3-kinase (PI3K); protein kinase A (PKA); protein kinase C (PKC); and c-jun N-terminal kinase (JNK). Retrograde trafficking systems transport these to the cell soma which then activates transcription factors such as CREB (cAMP responsive element binding protein), c-Jun, and ATF-3 (activating transcription factor 3). Combination of these processes enable activation of regeneration associated genes (RAGs) converting the nerve from a neuro-transmitter state to a regenerative state. For instance, downstream of the injury-induced PI3K-GSK3 (glycogen synthase kinase-3 signaling pathway), upregulates and phosphorylates SMAD family member 1 (SMAD1), causing its accumulation in the nucleus where it interacts with histone acetyltransferase p300 (p300HAT) which causes the over-expression of genes encoding cytoskeleton proteins such as tubulin and actin. Additionally, transcription factors (Sox11, c-JUN); cell adhesion and guidance receptors; growth-associated protein-43 (GAP43); and neurotropic factors (NTFs) such as NGF (Nerve Growth Factor) and BDNF (Brain-Derived Neurotropic Factor) and their receptors Trk, are also upregulated. There is a simultaneous downregulation of neurofilaments, neuropeptides and neurotransmitters gene expression, thus inhibiting the neurotransmitter function of the neuron. It also results in the ‘cell body response’ that is seen after PNI and includes swelling of the neuronal cell; chromatolysis (scattering of Nissl bodies or rough endoplasmic reticulum); migration of the nucleus to the peripheral of the neuron; and increased protein synthesis for regeneration [7,8,9,10,11,12,13,14,15].

Distally, the axons and myelin in the distal stump degenerate and are cleared (via a cascade of molecular and cellular events known as Wallerian degeneration (WD) (Figure 2) [16,17]. WD is a crucial process as the debris generates inhibitory signals such as myelin-associated glycoprotein that prevent axonal nascent neural growth from the proximal stump. SCs play a key role in the distal segment during WD. Axonal degeneration causes the loss of axonal contact with their SCs causing these adaptable cells (i.e., SCs) to re-enter the cell cycle and de-differentiate from a supportive to a repair phenotype. Gene expression of structural proteins such as myelin basic protein and myelin-associated glycoprotein are reduced, while reparative proteins such as NGF, basic fibroblast growth factor, Neurotrophin-3 (NT-3), neural cell adhesion molecule (NCAM), and glial maturation factor-β gene expression is upregulated. The de-differentiated SCs phagocytize degenerated myelin and repurpose it for future remyelination. They also express chemokine C-C motif ligand 2 (CCL2), leukemia inhibitory factor (LIF), IL-1α, IL-1β, IL-6, and pancreatitis-associated protein III (PAP-III) through the leaky nerve-blood barrier to recruit immune cells for further debris clearance [13,14,15,16,17]. Among immune cells, the first responders to the injured area are neutrophils, followed by macrophages that arrive 2–3 days after injury and peak at one week [16]. Macrophages clean up myelin via phospholipase A2 (PLA2) initiated phagocytosis, whose products, lysophosphatidylcholine (LPC) and C-reactive protein, activate the classic complement pathway [16]. In addition to cleaning up the debris, macrophages contribute to neural regeneration by producing proteases and growth-promoting factors, stimulating ECM remodeling, and regulating SCs. In addition to cleaning up the debris, macrophages contribute to neural regeneration by producing proteases and growth-promoting factors, stimulating ECM remodeling, and regulating Schwann cells [16].

## 3. End-to-End Repair

Early peripheral regeneration studies by Cruikshank in the late 18th century were instrumental in the ultimate surgical repair of nerves. Von Langebeck performed one of the earliest successful surgical nerve repairs of the median nerve in 1876, with return of function in a year [2,18,19]. Woolsey was first to describe the end-to-end suture technique in 1907 which led to the end-to-end (ETE) surgical repair being more acceptable in clinical practice [18,19]. World War I accelerated the developments in the field of nerve repair and yielded procedures such as nerve grafting, described by Tinel and Elsberg [16]. As nerve grafting became more popular, Tinel, Elsberg, and Babcock detailed surgical decision making and when end-to-end repair is preferred to grafts [16]. In 1964, Smith popularized the use of the microscope for peripheral nerve surgery, which enabled more advanced nerve repairs and led to the modern surgical nerve repair techniques [18,19]. 

Primary end-to-end neurorrhaphy (ETE), is the current standard for nerve repair if the repair can be performed in a tension-free manner [18]. If a tension-free primary repair is not possible a nerve graft (autograft vs. allograft) is utilized to bridge the gap and coapted to the proximal and distal nerve stumps using the ETE technique. Several end-to-end repair techniques have been attempted, including epineural, fascicular, and group-fascicular repair (Figure 3) [2]. Epineurial repair involves placing microsutures through the epineurium without traumatizing the fascicles. It has the advantage of reducing suture material between the two ends and therefore does not hinder axonal regeneration. It is also technically easier to perform and does not damage axons. Fascicular repair attempts to minimize the misdirection of fibers by allowing for direct coaptation of fascicles, but is technically extremely challenging, requires a greater number of microsutures through the perineurium, and has a higher chance of axonal damage. The additional sutures also tend to cause more foreign body reaction and scarring, hindering axonal regeneration. Group-fascicular repair entails repair of groups of fascicles rather than single fascicles and reduces fascicular injury and scarring, however it still creates additional tissue damage and foreign body reaction with additional sutures when compared to epineural repair [2,3]. Thus, the epineural repair (Figure 3A) has emerged the current gold standard for ETE nerve coaptation since it has the least interfascicular foreign body reaction, least scaring and best functional outcomes of these three techniques [2,18,19,20].

Timing and distance of the nerve injury from the target muscle are extremely important considerations while repairing peripheral nerve injuries. In general, the optimal timing for performing nerve repair is generally accepted to be within 72 h of injury, with acceptable outcomes up to 7 days after injury [2,21]. There are three time-dependent events that occur after nerve injury: (1) Wallerian degeneration; (2) the rate of nerve regeneration; and (3) motor end plate (MEP) degeneration that occurs after prolonged denervation. Axonal regeneration only commences once Wallerian degeneration is completed which takes 3–4 weeks to be completed. Once WD is complete, axonal regeneration begins at the rate of 1 mm/day or 1 inch/month [13]. Additionally, MEPs irreversibly degenerate at 12–18 months after injury if not reinnervated by then, due to the loss of nerve stimulation. Hence with proximal nerve injuries that are over 18 inches proximal to the target muscle (for instance neck injuries or brachial plexus injuries), even if the nerve coaptation is performed immediately after injury, it is destined to fail since the regenerating axons will only reach the MEPs after 18 months, by which time the MEPs have already irreversibly degenerated. Similarly, if the repair is performed after 12–18 months, there would not be any effective motor functional recovery, making it imperative to repair nerve injuries as soon as possible, especially in proximal injuries [2,13,20,21]. Hence for proximal injuries or delayed presentations, nerve transfers are utilized wherein a donor nerve (from a less useful synergistic muscle) closer to the denervated (but more useful) muscle is coapted primarily (i.e., ETE) to rescue denervated MEPs. This does require the sacrifice of a healthy donor motor nerve, which denervates a healthy, albeit less useful muscle. Sensory nerves are far more forgiving with regards to return of function after delayed or proximal repairs. ETE nerve coaptations are the most utilized surgical technique for PNIs clinically, however, there are several important issues to consider while employing this technique. ETE works best with a tension-free anastomosis, i.e., the proximal and distal end can be brought together without any tension on the nerve. Additionally, it needs a proximal nerve stump to be available, which as discussed above, is not possible for nerve avulsion injuries where the nerves are avulsed from their nerve roots in the spinal cord.

In end-to-end repairs, the injured axon in the proximal nerve segment elongates giving rise to axonal sprouts with a highly mobile tip known as the growth cone. The growth cone can assume different shapes to survey their environment via membrane protrusions called filopodia and lamellipodia [20]. Growth cones guide the axons into the distal nerve segment along the gradient of neurotropic cues in the microenvironment, including extracellular matrix, guidance receptors, cell adhesion molecules and NTFs [21,22]. Axonal growth thus occurs in three stages: protrusion, driven by filamentous actin (F-actin); engorgement, led by microtubule transport of organelles into the region; and consolidation, which occurs when the proximal growth cone is stabilized [23,24]. This process is tightly governed by molecular mechanisms that we explore in this section and are summarized in Figure 4. As previously mentioned, neuronal injury triggers a series of signals that ultimately alter gene expression within the injured neurons [11]. One mechanistic component involves the downregulation of myelin-differentiation genes: myelin transcription factor Egr2 (Krox20), cholesterol synthesis enzymes, P0 structural protein, myelin basic protein (MBP), and membrane associated proteins, such as myelin associated glycoprotein (MAG) and periaxin [25,26]. These steps are important to remove inhibitory signals, allowing the axonal sprouts to reach their distal sites. Subsequently, genes that are involved in the recruitment of pre-myelinating Schwann cells are up-regulated, notably L1, neural cell adhesion molecules (CAMs), p75 neurotrophin receptor (p75NTR), and glial fibrillary acidic protein (GFAP) [25,26,27]. Upregulation of RAGs (also known as growth-associated genes) in SCs is transient but is another essential step that includes secretion of NTFs, such as BDNF, glial derived neurotrophic factor (GDNF), nerve growth factor (NGF), artemin (Artn), neurotrophin-3 (NT3), VEGF, and ciliary neurotrophic factors (CNTF) (Figure 4) [11,26,27,28,29,30]. Of note, CNTF promotes phosphorylation of transcription factor STAT3 which is retrogradely transported to the neuron soma and is essential for axonal regeneration and neuronal protection against apoptosis. In addition to neurotrophic factors, cytoskeletal proteins undergo qualitative and quantitative changes characterized by an increase in actin, peripherin and tubulin that are essential for growth cone adhesion, and a decrease in neurofilaments. Basal lamina proteins, such as laminin allow SCs to interact with the growth cone adaptor molecules to promote axonal growth into the endoneurial tubes of the distal nerve stump following end-to-end repair [7,12,31,32]. Clinically, the IKVAV motif of laminin has been used in stem cell therapy for peripheral nerve injury due to its ability to guide SC migration [33,34,35]. Changes in SC gene expression are controlled by factors like c-Jun, which induces a cascade involving a transient upregulation of cyclin-dependent kinase Cdc2, in turn phosphorylating vimentin and allowing it to interact with basal lamina proteins [12,36,37,38,39]. Guidance molecules such a Netrin-1, Ephrin B2 and Slit3 are also upregulated at the injury site within the fibrous bridge that forms across the coaptation site after ETE coaptation. Macrophages secrete high levels of Slit3, while fibroblasts within the bridge and the advancing axons secrete Robo1 receptors their surface. The Slit3-Robo1 signaling axis acts as a restrictive signal to keep the axons within the nerve bridge. Netrin-1 expression on SCs has a dual action on endothelial cells (ECs) and the growth cone. It interacts with CD146 receptors on ECs, boosting their proliferation and migration, while it binds with DCC receptor (Deleted in Colorectal Cancer) on advancing axons, guiding the growth cone across the injury site [35].

Thus, extracellular matrix proteins, NTFs, and RAGs expressed by SCs are crucial for successful axonal growth, and if any component is missing, regeneration will not occur [22]. Notably, these changes in cellular expression are transient, and declines further if neurons are not in contact with their target, the distal segment [16]. This has significant clinical implications. Firstly, it explains why surgical nerve coaptations need to be performed within 72 h if possible. Secondly, it is essential to freshen the nerve ends prior to coaptation so as to prime the neurons to express the optimal RAGs. Freshening the nerve endings also helps tip the balance between the M1 and M2 macrophage polarization towards nerve regeneration. A shift towards M1 state and a lack of M2 macrophages leads to chronic inflammation and impaired nerve regeneration. M2 macrophages are extremely important to nerve regeneration due to their ability to response to injury-induced hypoxia. Hypoxia triggers hypoxia-inducible factor 1 alpha (HIF-1α) production in macrophages which is followed by an increased expression of VEGF-A (Vascular Endothelial growth factor A). This in turn increases EC migration and angiogenesis across the coaptation site, that guide SCs across to form bands of Büngner across the injury site. These bands are crucial for axonal guidance as axons march across the gap from the proximal to the distal nerve segment [16].

## 4. End-to-Side Nerve Coaptation

End-to-side nerve coaptation (ETS) is a surgical repair technique in which the distal stump of a transected nerve is attached to the side of an adjacent nerve, with or without the creation of an epineural window (Figure 5) [34]. The basic concept of ETS is based on axonal regeneration along the distal stump of a transected nerve by inducing collateral axonal sprouting from a neighboring healthy nerve. This procedure is most often indicated in cases where the proximal stump of an injured nerve is unavailable (such as nerve root avulsions) or if the resulting gap is too large to be bridged by a nerve graft. ETS was first introduced by Letievant in 1873 and was described as a reconstructive strategy for peripheral nerve injuries with extensive tissue loss [34,35]. The first published reference of end-to-side nerve coaptation was published in 1903; however, it was not until Viterbo et al. reintroduced the technique in 1992 that it once again piqued the interest of the surgical community and became a clinically utilized procedure [35]. Currently, this procedure is used to treat a variety of nerve injuries, including those involving large gaps (50 mm), the brachial plexus nerve root avulsion injuries, facial nerve palsy, and in the treatment of painful neuromas [34,35,36]. With respect to brachial plexus procedures, the first reported case was performed in 1995, when Piennar et al. sutured the C5 and C6 nerve roots to the phrenic nerve. Viterbo performed a cross-facial nerve graft connecting the buccal branches and temporozygomatic branches utilizing the ETS technique in 1993 and successfully reinnervated the facial muscles [5,36]. Al-Qattan applied this technique to painful neuromas of the superficial branches of the radial nerve, achieving successful resolution in all eight patients. Aszmann et al. confirmed favorable results in 16 out of 17 patients who were affected by painful neuromas of sensory nerves in the upper and lower limbs [34]. ETS has several advantages compared to ETE: (1) technically easier and faster to perform; (2) there is no sacrifice of a donor nerve required and therefore no functional deficits; and (3) since the donor nerve is not sacrificed, it is easier to find a donor nerve by finding a functioning motor branch closer to the target muscle, theoretically making reinnervation faster (i.e., closer to the target MEPs, faster the reinnervation) [6]. However, studies have shown that regeneration is faster through ETE than ETS repairs, likely because the end-to-end technique does not require collateral sprouting, saving approximately two months of regeneration time [6]. Additionally, Sanapanich et al. found that ETE repair yields a higher number of axons that are less scattered with thicker myelin and have a greater muscle strength recovery [23]. This explains why ETE repair is clinically favored over ETS repair. That said, the ETS repair is still a valuable tool in the armamentarium of peripheral nerve surgeons and understanding the molecular mechanisms involved is crucial during surgical decision making.

Viterbo et al. demonstrated histologic evidence of axonal growth into the distal recipient nerve after ETS and electrophysiologic evidence of reinnervation, proving that axonal growth does occur at the coaptation site [36]. This phenomenon however lends itself to two interesting questions, i.e., (1) where exactly do the donor axons arise from, and (2) should an epineurial versus a perineurial window be made in the donor nerve to initiate collateral axonal sprouting. Despite multiple experimental studies on ETS nerve repair, the origin of regenerating axons remains debatable. Early studies suggested that regenerated axons arose from terminal sprouting, i.e., from the ends of damaged axons in the donor nerve. This theory therefore encouraged opening an epineurial window and intentionally damaging donor axons [38]. However, ETS axonal regeneration can occur even without opening an epineurial window and without sutures by simply gluing the end of the recipient nerve to the side of the donor nerve [40,41]. The current concept therefore is that ETS coaptation leads to collateral sprouting that occurs from the nodes of Ranvier closest to the coaptation site, along the length of the uninjured donor nerve and into the recipient nerve (Figure 6) [40,41,42,43,44]. This leads us to the second question, if ETS axonal regeneration can occur without an epineurial window, is the epineurial window really necessary? Hayashi et al. compared atraumatic vs. traumatic (epineurial window) of ETS and noted that the atraumatic ETS may produce modest sensory collateral sprouting but it is delayed and insufficient to be of any clinical consequence. Axotomy on the other hand produced a more robust axonal sprouting and was necessary for a clinically relevant functional recovery [45]. Noah et al. evaluated ETS coaptation in four groups: intact epineurium, epineurium window, perineurial window and partial neurectomy and noted that all groups demonstrated lateral sprouting, but the last two groups had the highest axonal count indicating that epineurium and perineurium form barriers for axonal regeneration in ETS [46]. These findings conflict with the work by Viterbo et al. who used a peroneal nerve transection rat model and showed that there is no significant difference in axon counts distal to the coaptation in groups with and without perineurial windows [47].Walker et al. studied the effects of perineurial window size on collateral axonal sprouting, blood-nerve barrier architecture and functional impairment of the donor nerve following small (1 mm) versus large (5 mm) perineurial windows in a rodent model of ETS between peroneal and posterior tibial nerve [44]. They found that while the larger (5 mm) perineurial window produced increased WD and worsened integrity of the blood-nerve barrier, functional recovery was not affected by the size of the perineurial window, indicating no advantage for a larger perineurial window. Overall, further studies need to be conducted with regards to the clinical efficacy of each of these ETS technique. However, the data so far seems to clearly indicate that an epineurial window at the very least is required while performing ETS. The superiority of a perineurial window in ETS coaptations is however inconclusive.

Previous sections have discussed the molecular mechanism for end-to-end nerve coaptation and the significance of SCs, RAGs, NTFs, and extracellular membrane proteins. It is possible to extend many of these concepts to ETS coaptation, while also exploring some of the unique nuances of this repair. For instance, the epineurial window besides removing the physical barrier, also produces a local injury to the donor nerve, producing a loss of axonal contact leading to switching local SCs from supportive to the repair phenotype. It also produces local “healthy” inflammation leading to macrophage recruitment and increase in NTFs (Figure 6) [37]. Lykissas et al. showed that prior to axonal regeneration, de-differentiated SCs are organized into columns at the coaptation site, providing channels through which axons regenerate. The cells will eventually invade the epineural layer of the coapted recipient nerve end (Figure 6) [37]. Zhang et al. utilized a fluorescent double-labeling technique to explore ETS nerve coaptation, and confirmed collateral sprouting from the donor nerve, but noted that SCs also migrate from the recipient nerve ending to the coapted donor nerve in addition to those from the donor nerve, together these SCs stimulate collateral axonal sprouting (Figure 6) [39]. Macrophage assisted elimination of myelin debris, production of growth-promoting factors, and stimulation of extracellular matrix remodeling enhance the process of nerve regeneration. In addition to these general molecular changes, many specific growth factors and receptors have been found to influence nerve regeneration at the site of ETS coaptation. For instance, NT-3 and its receptor Trk C; BDNF and its receptor Trk B are upregulated at the ETS coaptation site [37]. Additionally, growth-associated protein-43 (GAP-43), a marker of growth cone formation is also upregulated [24,35,40,41,42,43,44,45,46,47]. Finally, NGF and CNTF are upregulated and been shown to be crucial in promoting axonal regeneration following ETS [29]. Although these above-mentioned growth factors are necessary for nerve repair regardless of surgery type, there are subtle differences in ETS vs. ETE. For instance, NGF is more important for axonal branching making it crucial in ETS, while GDNF is more important for axonal elongation, making them vital in ETE repairs [25,32]. Thus, delineation of specific factors’ influence on various nerve repairs may have important clinical implications and further studies to elucidate the temporal, spatial and technique specific expressions are required to optimize the ETS technique.

To complete the discussion of the ETS technique, it is necessary to discuss a more recent technical innovation known as the supercharged end-to-side (SETS) transfer. Here the proximally injured nerve is repaired primarily and supercharged with an ETS coaptation form a donor nerve closer to the target muscle and distal to the primary repair site. The ETS coaptation distal to the injury site acts as a “baby-sitter” procedure, whereby the SETS donor axons rapidly innervate and support the denervated muscle and keeping the MEPs from disintegrating until the native axons can regenerate and travel down the injured nerve [48]. In other words, the SETS provides additional and early reinnervation of the MEPs, babysitting them, until the native axons can reinnervate them, thus mitigating the effects of chronic denervation. It has the additional advantage of not requiring the sacrifice of a donor nerve. SETS also induces a second site (the first being the primary repair site) for SC proliferation and upregulation of NGFs. This in turn prevents SC senescence since prolonged periods without axonal contact causes SC atrophy and cause them to lose their ability to maintain the bands of Büngner [49,50]. Preclinical models using histology, immunofluorescence and retrograde labeling, have demonstrated robust axonal regeneration across coaptation sites [50,51]. Lieu et al. demonstrated that an upregulation of Insulin-like growth factor (IGF-1) and its receptor (IGF-1R) is upregulated after SETS, and are critical for the survival, proliferation, and differentiation of sensory and motor axons [52]. They also demonstrated a reduction of TNF-like weak inducer of apoptosis (TWEAK), which is an important mediator of denervation-induced skeletal muscle atrophy [52]. The use of Thy1-GFP rats showed penetration of donor axons across the SETS coaptation sites in a tibial nerve model by day 7 and diffuse spread across the entire cross-sectional area of thee nerve by day 10 [49]. Furthermore, the axonal growth was noted to be both in an antegrade and retrograde direction in the recipient nerve. There was no difference noted in the number or size of axons proximally versus distally [53,54]. The data so far demonstrates a consensus that robust donor axon regeneration occurs across the SETS coaptation site and that these donor axons despite their bidirectional growth upon entering the recipient nerve are capable of reinnervating MEPs and reducing muscle atrophy. However, the impact of donor axons on the regeneration of native axons from the proximal repair site is still unknown. The relative contributions of donor and native axons to functional outcomes and whether these relative contributions change over time is also unknown. Finally, the extend of the donor nerve window in terms of length and depth (i.e., epineurial vs. perineurial) has also not been fully elucidated. Thus, despite the apparent clinical advantages of maintaining muscle functionality for very little clinical disadvantage i.e., minimal donor nerve loss of function, the clinical adoption of this technique has been slow.

## 5. Side-to-Side Nerve Coaptation

Despite the many advancements in peripheral nerve repair been made in the late 20th century such as the ETE and ETS techniques, several limitations remained including the sacrifice of healthy donor nerves and the extended nerve regeneration time in high level injuries, with resultant MEP degeneration. In an attempt to address some of these pitfalls, Zhang et al. developed the side-to-side neurorrhaphy technique (STS) in 1994 [39]. The STS approach involves the coaptation of the side of the injured nerve, distal to the site of injury, to a healthy, intact donor nerve through the creation of epineural windows. This may be accomplished either through direct coaptation or through one or more allograft or autograft cross bridges [39]. This method has been used both in isolation and in conjunction with the other repair techniques of ETE and ETS (Figure 7). There has been a combination of multiple animal studies and a smattering of human studies, published on this relatively new method [55,56,57,58,59]. In 1999, Yuksel et al. performed side-to-side nerve coaptation procedures between the peroneal and tibial nerve trunks of ten rats and demonstrated that axonal passage was possible with side-to-side nerve neurorrhaphy. The functional results were satisfactory and were found to be superior to the end-to-side nerve coaptation technique [55]. Approximately 17 years later, in 2016, Reichert et al. examined side-to-side nerve coaptation as a means of left limb brachial plexus repair in six New Zealand white rabbits [56]. The ventral branches of the C5 and C6 spinal nerves were avulsed from the spinal canal. The side-to-side nerve coaptation was performed using a 3-mm window on the right side of the C5 and C6 spinal nerve ventral branches and a 3-mm window on the left side of the C7 spinal nerve ventral branches [56]. The control group for this study was six healthy right limbs of the white rabbits. The C5 and C6 nerves were harvested 1 cm proximal and distal to the coaptation sites and compared to the control group nerves from the same region and an axonal count and G-ratio calculated. The G-Ratio is the ratio of the axon diameter to the diameter of the entire fiber and is indicative of normal conduction in myelinated fibers. They found that the number of myelinated axons was markedly reduced in the side-to-side groups compared to the control group, but the experimental group had a G-Ratio of > 0.6 indicating a functional transfer of axons after STS coaptation [56]. In 2012, Zhang et al. evaluated the clinical outcomes of side-to-side neurorrhaphy in twenty-five patients who had sustained high-level peripheral nerve injuries. They noted a functional recovery rate of 60% for motor function and 100% for sensory function. They concluded that this technique is a promising and feasible method in high peripheral nerve injury repairs [39]. Overall, the use of STS repair, both alone and in conjunction with primary ETE repair, remains infrequent likely due to the fact that the exact molecular mechanism of how regeneration occurs in the absence of direct axonal injury remains unclear. One possible explanation has been proposed by Napoli et al., who developed a mouse model in which the activation of the Ras-MAP signaling pathway was sufficient to initiate SC dedifferentiation without axonal injury [56]. In this pathway, Ras activates the proto-oncogene serine threonine protein kinase Raf, which then activates mitogen-activated protein kinase-kinase (MEK), in turn promoting the mitogen-activated protein kinase (ERK) signaling that maintains the dedifferentiated state of SCs (Figure 8) [58,59,60].

Overall, STS repairs show variable nerve regeneration and axonal counts, which are insufficient for optimal nerve repair by itself. Instead, STS repair can serve as a temporary ‘baby-sitter’ of the denervated motor end plates, in proximal nerve injuries (similar to the SETS technique noted above), until the native axons regenerating from the proximal primary ETE repair site can reinnervate them, thus mitigating the effects of chronic denervation (Figure 7C). It has the additional advantage of not requiring the sacrifice of a donor nerve and induces a second site for SC proliferation and upregulation of NGFs, in turn improving native axonal regeneration. Of note, unlike SETS, STS has an added advantage of being able to be performed at multiple locations along the length of the nerve (Figure 7D). Shea et al. demonstrated that a side-to-side bridge consisting of a collagen based nerve conduit coupled with an ETE repair of the proximally transected nerve resulted in less muscle atrophy and fewer signs of muscle denervation, such as pyknotic nuclei and smaller muscle fibers, than ETE repair or STS bridge alone [54]. The authors explained that a benefit of using a bridge is to obviate the extensive nerve dissection required to mobilize the nerves to create a direct STS connection thus avoiding inadvertent injury to the nerves during dissection [58]. The molecular mechanism of STS bridges was investigated by Gordon et al., who found that they help sustain the donor segment by releasing growth-supporting factors such as cAMP and neuregulin from the donor nerve [60]. These factors sustain the SCs until native axons can take over. However, similar to SETS, the impact of donor axons on the regeneration of native axons from the proximal repair site; the relative contributions of donor and native axons to functional outcomes; and whether these relative contributions change over time is also unknown. Finally, the extend of the donor nerve window in terms of length and depth has not been optimized, and further studies are necessary to further elucidate the molecular mechanisms of this repair before a wider clinical use of this technique is to be expected [61]. 

## 6. Muscle Reinnervation

In normally innervated muscles, motor branches (motoneurons) do not branch until they reach their destined muscle. Once intramuscular, they branch out with each motoneuron innervating a discrete territory in the muscle cross-section in the form of a ‘mosaic’ pattern [62]. Each of these functional units are known as a motor unit. With partial nerve injuries, the mosaic pattern of distribution is replaced by a “clumping” pattern, in an inverse ratio, i.e., lower the number of intact Mus, higher the clumping pattern. After complete nerve injury and end-to-end repair and reinnervation, the normal mosaic pattern is completely replaced by a “clumped” distribution. This occurs because of two reasons: firstly lower number of motoneurons are reaching the muscle and therefore compensate by capturing a higher number of muscle fibers in the vicinity with a resultant increased size of each MU, and a clumped pattern of muscle capture. Secondly since perisynaptic SCs present at the endplate region of the neuromuscular junction help guide regenerating axons to the end plate after nerve repair, regenerating axons miss “branch points”, being guided to the motor end plates by neutrophic factors generated by these presynaptic SCs. Thus, branching occurs only after they reach the muscle, accounting for the “clumping’ pattern. In general there is a 5–8 fold limit to this compensation mechanism, i.e., each motoneurons can only capture 5–8 fold more denervated muscle fibers. Thus, when less than 20% motoneurons reinnervate the denervated muscle, the entire muscle cannot be recaptured and muscle force declines [63]. Lack of muscle reinnervation can also occur secondary to misdirection since most developmental axon guidance cues are lost in adults. Axons can randomly enter vacant endoneurial tubes distally with sensory to motor and vice versa cross innervation. Alternatively motoneurons reinnervate different muscles than their native muscle [7]. 

Henneman’s size principle describes the relationship between the size of the motoneurons with the muscle fibers they innervate i.e., motoneurons with large cell bodies innervate fast-twitch, less fatigue-resistant muscle fibers, whereas motoneurons with small cell bodies innervate slow-twitch, higher fatigue-resistant fibers. These size relationships are lost after nerve transection and repair but return after axonal regeneration and muscle reinnervation. The size of nerves proximal to the site of transection and repair decreases after axotomy. Denervated muscle fibers also atrophy prior to their reinnervation. As reinnervation occurs, muscle fibers increase in size and nerve fibers recover their normal size. However, since the regenerating axons capture new muscle territory, it is the size-dependent branching of the regenerating nerves in the denervated intramuscular nerve sheaths and therefore the increase in MU size rather than then the cell body size that restores the MU forces and Henneman’s size principle [7]. 

As mentioned in above sections, after nerve injury, motoneurons undergo early and late changes in gene expression switching them to a regenerative pheonotype [63]. One intriguing aspect of this response is the intense shedding of synapses especially those of glutamatergic origin from motor cell bodies in the ventral horn of the spinal cord. Alvarez et al. distinguished this neuronal plasticity of motoneurons into two types: (1) a faster transient loss of synapses over the cell bodies of axotomized motoneurons that affects all types of synapses. And (2) a slower but permanent change in spinal cord circuitry that not only permanently affects axotomized motoneurons but other targets in the ventral horn, affecting both cell bodies and dendritic arbors, thus reconfiguring ventral horn motor circuitries to function after regeneration without direct proprioceptive or sensory feedback from muscle [63]. This process is modulated by injury severity suggesting a correlation between poor regeneration with sensory and/or motor targeting errors in the periphery that would render the previous central circuitries non-functional. Peripheral targeting errors thus must necessarily scramble motor circuit organization in the spinal cord rendering them dysfunctional. Thus, after nerve regeneration, the ventral horn operates without feedback about muscle length or motor output causing permanent changes in motor function. The extent to which the loss of these synaptic inputs further worsens peripheral function versus optimizes central circuits to the vagaries of jumbled peripheral targeting errors is unknown and currently under investigation.

## 7. Conclusions

With the high prevalence of peripheral nerve injuries in the United States and the variety of techniques that have been developed to treat individuals suffering from such injuries, it is essential to better understand the benefits and drawbacks of the available treatment strategies, especially of their individual molecular mechanisms. In cases where the surgical gold standard of tension-free direct end-to-end repair is not possible, autograft or allografts are utilized in repairing the nerve graft. If the injury is too proximal or if primary repair is not possible as in the case of nerve root avulsion injuries, it is possible to employ alternate and/or adjunct reconstructive strategies such as ETS, SETS and STS nerve coaptations. Although each of these methods has a unique clinical indication, there is significant overlap between the molecular mechanisms of axonal regeneration for ETE, ETS, SETS, and STS nerve coaptation. Specifically, the process of Wallerian degeneration, SC de-differentiation, chemoattraction of macrophages, expression of NGFs and RAGs are crucial for all three coaptation techniques to be successful, with some subtle variations in each coaptation. These variations have not been thoroughly understood and are potential areas of further studies that in turn could open up opportunities for adjunct therapies to augment nerve regeneration. While rodent models are an established animal model for peripheral nerve injuries due to their size and cost effectiveness, they have several disadvantages that must be acknowledged. For instance, rodent peripheral nerve regeneration is far more rapid than human nerve regeneration and is not clinically translatable. Additionally, it is difficult to distinguish epineurium from perineurium since the two layers are often confluent, making it challenging in studies to differentiate the effects of each of these on outcomes. Additionally due to the miniscule size of the nerves, it is possible to damage axons in the donor nerves, leading to an inadvertent ETE coaptation with donor nerves in ETS or STS coaptations, thus confounding the results. This is likely the reason for the variable results we noted in the ETS, SETS and STS sections above regarding epineurial vs. perineurial windows. Current gaps in knowledge will likely require large animal studies to elucidate information regarding the effects of “baby-sitter” procedures such as SETS or STS such as the optimal length and depth of coaptation and their effect in priming the neuronal soma. The effect of the donor axons on native axonal regeneration and the relative contributions of donor and native axons to functional outcomes and whether these relative contributions change over time needs further study. Understanding these molecular mechanisms is important for us to better optimize surgical decision making and improve outcomes in complex nerve injuries.

## Figures and Tables

**Figure 1 jcm-12-01555-f001:**
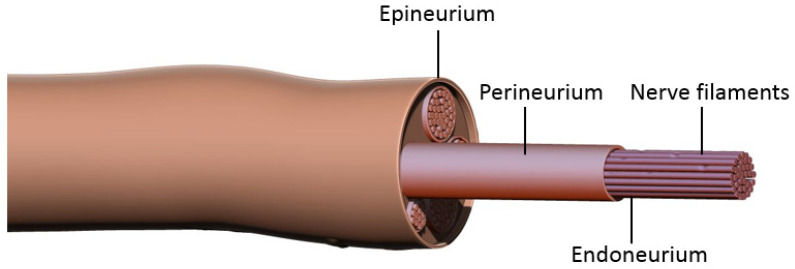
Peripheral nerve structure. Each axon is enveloped by endoneurium and Schwann cells. Groups of these nerve filaments are organized into fascicles by perineurium, and these fascicles are finally sheathed in epineurium to form a peripheral nerve.

**Figure 2 jcm-12-01555-f002:**
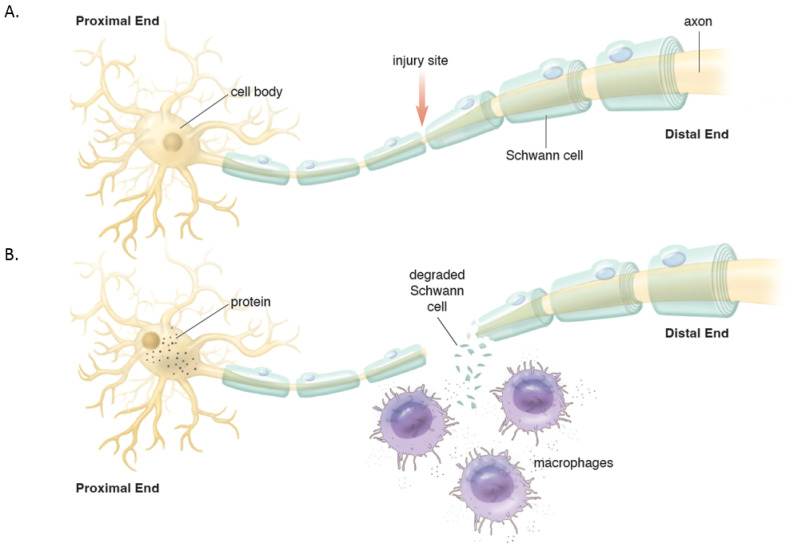
(**A**). Intact peripheral nerve. (**B**). Nerve transection leads to events in the proximal stump and distal stump. Proximally, the cell nucleus moves to the periphery and Nissl bodies disperse with increased protein synthesis to help repair the damage and seal the proximal membrane. Distally, Wallerian degeneration occurs which includes de-differentiation of Schwann cells, recruitment of macrophages that help breakdown and clear the debris in preparation of axonal regeneration.

**Figure 3 jcm-12-01555-f003:**
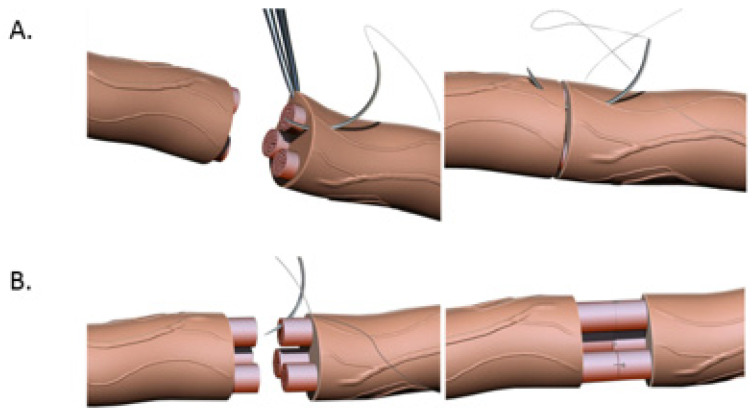
(**A**). Nerve coaptation via epineural suturing technique demonstrating sutures in the epineurium. (**B**). Nerve coaptation via fascicular repair demonstrating sutures in individual fascicles.

**Figure 4 jcm-12-01555-f004:**
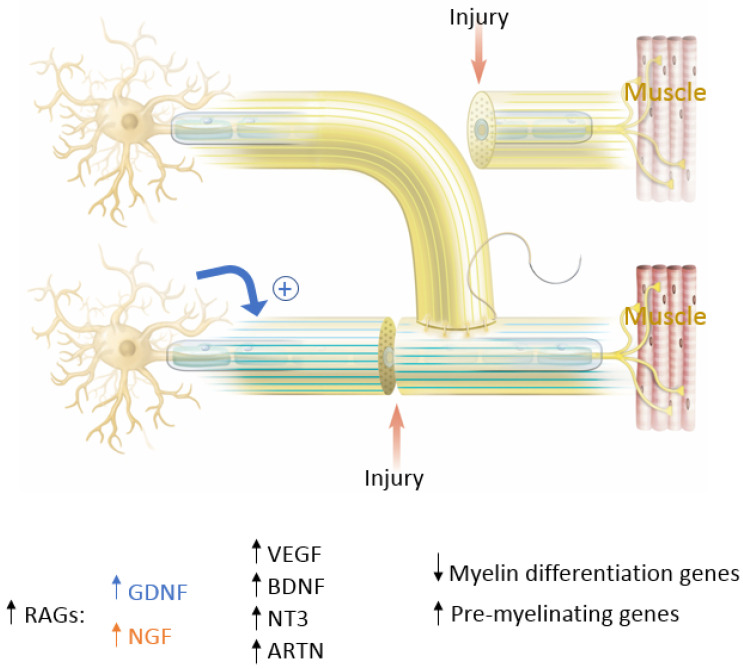
The key molecular players in peripheral nerve regeneration. In blue are factors that favor End-To-End (ETE) repair and in orange are factors that favor End-To-Side (ETS) repair. Factors that favor both ETE and ETS repair are in black. RAGs= Regeneration Associated Genes; GDNF = Glial Derived Neurotrophic Factor; NGF = Nerve Growth Factor; VEGF = Vascular Endothelial Growth Factor; BDNF = Brain-Derived Neurotropic Factor; NT3 = Neurotrophin-3; ARTN = Artemin.

**Figure 5 jcm-12-01555-f005:**
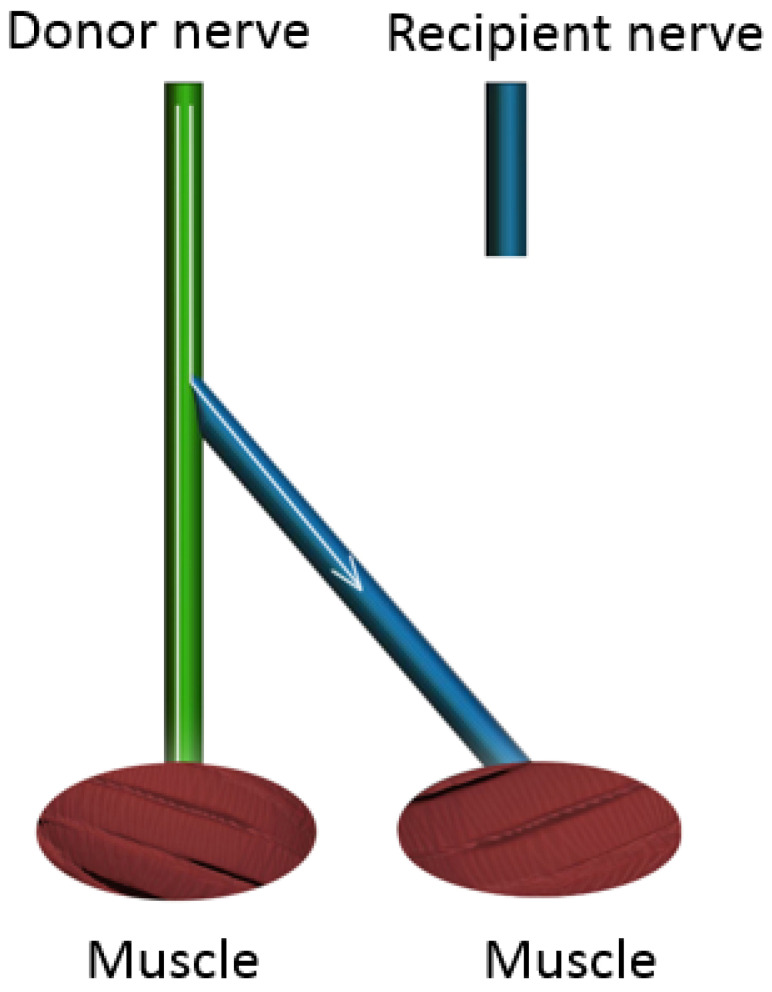
End-to-side nerve repair. Traditional end-to-side nerve coaptation involves the coaptation of the distal denervated stump into the side of the intact donor nerve. Any axons that are present in the distal recipient nerve stump exclusively originate from the donor nerve.

**Figure 6 jcm-12-01555-f006:**
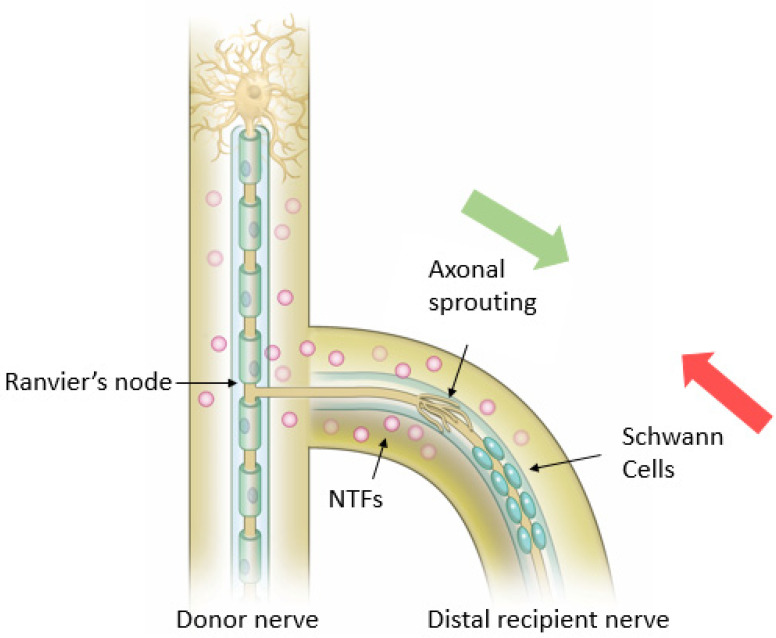
Proposed mechanism for axonal regeneration in End-To-Side (ETS) repair. Schwann cells from donor nerve and/or recipient nerve de-differentiate into the reparative phenotype and align themselves at the coaptation site. Collateral axonal sprouting occurs at the node of Ranvier closest to the coaptation site. NTFs = Neurotropic Factors.

**Figure 7 jcm-12-01555-f007:**
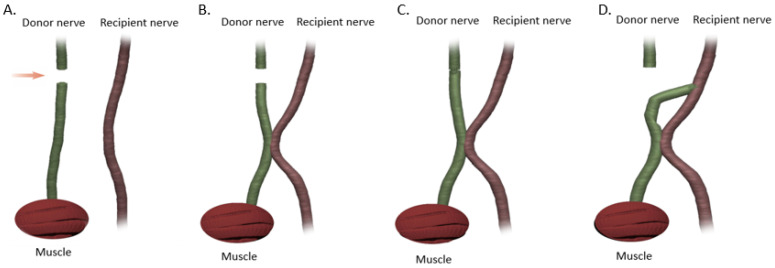
End-to-end (ETE) nerve repair technique demonstrating the coaptation of the injured nerve distal to the site of injury to an intact donor nerve (DN) through an epineural window. (**A**) Nerve injury. (**B**) Injury with downstream STS repair. (**C**) Injury with ETE repair at the site of injury and downstream STS repair. (**D**) Injury with ETS repair at the site of injury and downstream STS repair. Arrow points to the level of injury.

**Figure 8 jcm-12-01555-f008:**
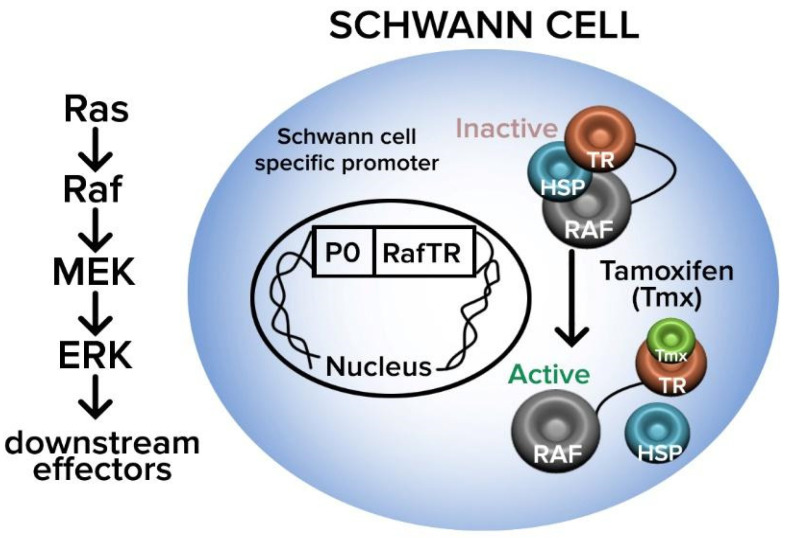
Ras-MAP signaling pathway for Schwann cell dedifferentiation as proposed by Napoli et al. Ras activates protein kinase Raf, which then activates mitogen-activated protein kinase-kinase (MEK), in turn promoting mitogen-activated protein kinase (ERK) signaling that then maintains the de-differentiated state of Schwann cells. TR = Tamoxifen-inducible RAF-Kinase; HSP = Heat Shock Protein; RAF = Rapidly Accelerated Fibrosarcoma (Adapted from Napoli et al. [60]).

## Data Availability

Not applicable.

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
