# Peer review of "Molecular Basis of Surgical Coaptation Techniques in Peripheral Nerve Injuries"

_jcm, 2023, doi:10.3390/jcm12041555_

Round 1

Reviewer 1 Report

Well written paper. 

I think it can be improved if the neuron and the end organ is included. E.g. how neuronal plasticity affects the outcome of the respective repair. What happens to size and number of muscle motor units after re-innervation and so on. 

nerve repair is mandatory but does not presuppose functional success. 

Author Response

Thank you for your comments. We have added an entire section on muscle reinnervation and synaptic plasticicity (Section 6, line 489-544)

Reviewer 2 Report

Excellent article.
Very well written, relevant, and important subject that is not always addressed by other articles in the same area.

Author Response

thank you for your comments, appreciate it. 

Reviewer 3 Report

This study aimed to summarize the similarities and divergences of the three commonly used nerve repair strategies and the scope of molecular mechanisms and signal transduction pathways in nerve regeneration.

The review is comprehensive in summary and fluent in line with the requirements of JCM. However, the references in the last three years are on the low side, and it is recommended to increase appropriately.

Author Response

Thank you for your comments, we have added 6 new references from within the last three years as suggested.